# Dietary Advanced Glycation End-Products (dAGEs) Intake and Bone Health: A Cross-Sectional Analysis in the Rotterdam Study

**DOI:** 10.3390/nu12082377

**Published:** 2020-08-08

**Authors:** Komal Waqas, Jinluan Chen, Bram C. J. van der Eerden, M. Arfan Ikram, André G. Uitterlinden, Trudy Voortman, M. Carola Zillikens

**Affiliations:** 1Department of Internal Medicine, Erasmus University Medical Center, 3015 Rotterdam, The Netherlands; k.waqas@erasmumc.nl (K.W.); j.chen@erasmusmc.nl (J.C.); b.vandereerden@erasmusmc.nl (B.C.J.v.d.E.); a.g.uitterlinden@erasmusmc.nl (A.G.U.); 2Department of Epidemiology, Erasmus University Medical Center, 3015 GD Rotterdam, The Netherlands; m.a.ikram@erasmusmc.nl (M.A.I.); trudy.voortman@erasmusmc.nl (T.V.)

**Keywords:** dietary advanced glycation end-products osteoporosis, fractures, bone mineral density, trabecular bone score

## Abstract

Animal studies suggest a role for dietary advanced glycation end-products (dAGEs) in bone health, but human studies on dAGEs in relation to bone are lacking. We aimed to study whether dAGEs intake is associated with the parameters of bone strength namely, bone mineral density (BMD), prevalent vertebral (VFs), and major osteoporotic fractures (MOFs = hip, wrist, proximal humerus, and clinical VFs). 3949 participants (mean age 66.7 ± 10.5 years) were included from a Rotterdam study for whom Carboxymethyllysine (CML—a dietary AGE) was estimated from food frequency questionnaires combined with dAGEs databases. Multivariable linear and logistic regression models were performed adjusting for age, sex, energy intake, dietary quality, physical activity, diabetes, smoking, renal function, and cohort effect and for models on fractures, subsequently for BMD. We observed no association of CML with BMD at both femoral neck (β = −0.006; *p* = 0.70) and lumbar spine (β = −0.013; *p* = 0.38). A higher intake of CML was linearly associated with VFs (Odds ratio, OR = 1.16, 95% CI (1.02–1.32) and a similar but non-significant trend with MOFs (OR = 1.12 (0.98–1.27). Additional adjustment for BMD did not change the associations. Our results imply a positive association between dietary intake of CML and VFs independent of BMD. Future studies are needed in order to elucidate whether associations found are causal.

## 1. Introduction

The impact of a healthy lifestyle, including diet, on bone health has been well established [1]. A healthy diet pattern rich in fruits and vegetables has been consistently associated with higher bone strength parameters and lower incidence of fractures [2,3,4,5]. In contrast, a diet rich in saturated fatty acids, meat, and processed foods has been associated with no or negative influence on bone health [6,7]. Previous studies have shown that such diets rich in fats, meat, and processed foods contain high amounts of dicarbonyl compounds (DCs), and their final by-product Advanced glycation end-products (AGEs). AGEs are a heterogeneous group of compounds that formed as a result of spontaneous glycation of amino groups of proteins, lipids, and nucleic acids both in vivo and in vitro [8,9]. Endogenously, the formation of AGEs is enhanced by aging and under the circumstances of oxidative and glycemic stress [10,11,12]. Exogenously, the major sources of AGEs are smoking and diet [13,14].

According to one estimate, an average human diet consists of approximately 1200 mg of DCs and 75 mg of AGEs per day [15]. DCs, such as methylglyoxal (MG), and AGEs, such as Carboxymethyllysine (CML), constitute a major proportion of our dietary AGEs (dAGEs) intake [14]. The amount of AGEs in Western diet is determined through macronutrient composition, along with processing temperature, time and moisture levels [16,17,18]. Once ingested, it remains uncertain how much of these AGEs and their precursor DCs would be absorbed through the alimentary canal into the circulation with reports varying from 30 to 80% [19,20,21]. From circulation, these AGEs eventually bind to proteins with long half-lives, such as collagen in bone. The accumulation of AGEs in bone has been associated not only with increased collagen cross-linking and stiffness [22,23], but also with the activation of inflammatory pathways and reduced bone turnover [24,25]. These mechanisms explain the implication of AGEs in reduced bone quality, biomechanics, and high fracture risk [26].

Multiple studies using models of animals fed on a diet whose AGEs content was high showed a negative impact on bone properties. In one study where rats were fed on bread crusts for 88 days, an increased accumulation of AGEs was found in tibia and reduced mechanical properties [27]. Illien-Junger et al. found that a high AGE-diet led to disturbed vertebral microarchitecture and reduced fracture resistance in young female non-diabetic, non-obese mice [28] when compared to young male and aged female mice. The scales of zebra fish fed a high fat diet showed a rise in circulating AGEs and impaired bone metabolism [29]. Together, these preclinical studies point to a role of dAGEs intake in reducing both cortical and trabecular bone properties.

Although the contribution of dAGEs to reduced bone mechanical properties and decreased fracture resistance has been observed in animal models [28,30], the role of dAGEs in human bone health has hardly been studied. An unstandardized dAGE quantification technique (ELISA vs. UPLC MS/MS) in current databases, heterogeneity of AGEs, and effect of different processing, cooking, and preserving techniques on AGEs content of a food item makes the study in humans challenging [14,17,31].

The aim of our study was to investigate whether dAGEs intake is associated with bone health measures, such as bone mineral density (BMD), and prevalent major osteoporotic fractures (MOFs) and vertebral fractures (VFs).

## 2. Materials and Methods

### 2.1. Study Population

Our cross-sectional analysis was performed in the participants of the Rotterdam study (RS), a population-based prospective cohort study. For the detailed design of the study, we refer to a recent update in 2020 [32]. Briefly, the participants were included at three different points in time, namely in 1990, 2000, and 2006 and named in ascending order as RS-I, RS-II (≥55 years), and RS-III (≥45 years) sub-cohorts based on year of inclusion, respectively. After inclusion visit, the participants were followed regularly every 4–6 years. The Rotterdam Study has been approved by the Medical Ethics Committee of the Erasmus MC (registration number MEC 02.1015) and by the Dutch Ministry of Health, Welfare, and Sport (Population Screening Act WBO, license number 1071272-159521-PG). The Rotterdam Study Personal Registration Data collection is filed with the Erasmus MC Data Protection Officer under registration number EMC1712001. The Rotterdam Study has been entered into the Netherlands National Trial Register (NTR; www.trialregister.nl) and the WHO International Clinical Trials Registry Platform (ICTRP; www.who.int/ictrp/network/primary/en/) under shared catalogue number NTR6831. All of the participants provided written informed consent to participate in the study and to have their information obtained from treating physicians.

For this particular analysis, we eventually included 3949 participants from RS-I (5th follow-up, *N* = 963 or 24%), RS-II (3rd, follow-up, *N* = 1250 or 32%), and RS-III (1st baseline visit, *N* = 1735 or 44%) with both available dietary data and bone parameters data obtained between years 2008–2012. We excluded all participants with missing data on effective glomerular filtration rate (eGFR), body mass index (BMI), smoking status, diabetes status, physical activity, and bone mineral density (BMD) (Figure 1).

### 2.2. Dietary Advanced Glycation End-Products (dAGEs) Assessment

#### Food Frequency Questionnaire

A self-administered semi-quantitative 389-item food frequency questionnaire (FFQ) was employed based on the category of foods consumed in the past one month in RS population between 2008 to 2012. Those 389-items have been categorized into 25 groups. Most of the groups are self-explanatory, as an example ‘sweets’, which included ice cream, chocolates, added sugar in tea and coffee, etc. [33]. The FFQ collects information on the food types, serving sizes, frequency of consumption, and sometimes preparation method. The FFQ has been validated in two other Dutch populations based on a nine-day dietary record and 4-week dietary history [34,35].

### 2.3. dAGEs Databases

Two published dAGEs databases have been used to obtain the reference amount of AGEs in a particular food item. Primarily, a Dutch database consisting of 190 food items for three AGEs, namely CML, methylglyoxal-derived hydroimidazolone (MG-H1), and carboxyethyllysine (CEL) consumed in current Dutch diet [31] and, secondarily, an Irish database consisting of 257 food items for only CML [36] have been used. Briefly, both of the databases made use of ultra-performance liquid chromatography tandem mass-spectrometry (UPLC–MS/MS) method for determination of protein bound-AGEs in frequently consumed food items in a Western diet.

In RS, we assume that the usual Dutch cooking methods would have been applied. Nonetheless, a difference in cooking time and temperature could not be taken into account when coupling to the dAGEs database. There were a few special scenarios during AGEs estimation:

If a food item was not found in both the reference databases, but on FFQ, a similar type of food item has been used.

If multiple AGE values are reported in dAGEs database for a single food item in the FFQ, then the mean of all values was used.

For a combination of food items, AGEs from individual food items were derived by using composition information on the package or a standard recipe online.

A detailed workflow from FFQ food item data to dAGEs has been previously published [37]. Briefly, daily intake of an AGE, such as CML, for one food item has been calculated by multiplying the CML value (mg/100 g) with the serving size (grams). Afterwards, the CML values from individual food items have been summed up to calculate the total intake of CML for a day (mg/day). Henceforth, we used energy-adjusted CML to minimalize the effect of total energy intake per day, which is calculated by using the residual method. We also performed similar calculations to estimate energy-adjusted MG-H1 and CEL for a secondary analysis although CML intake seems to be an ideal representative of dAGEs intake in relation to bone health.

A total dAGE score was calculated in three steps: firstly, the Z-scores were calculated for individual dAGEs (CML, CEL, and MG-H1) by using this formula = (individual dietary AGE minus total population mean dietary AGE/population standard deviation dietary AGE); secondly, an average of the three Z-scores representing Z-score of total dAGEs intake; and lastly, we adjusted this for energy intake to make a variable representing energy-adjusted total dAGEs intake.

Daily energy intake and macronutrient composition were calculated from Dutch food composition database (NEVO) [33]. The dietary quality was assessed as a diet quality score (0–14) reflecting adherence to Dutch dietary guidelines [33].

### 2.4. Measurement of Bone Mineral Density (BMD) and Trabecular Bone Score (TBS)

BMD was measured using iDXA Prodigy total body fan-beam densitometer (GE Lunar Corp., Madison, WI, USA) at two positions femoral neck (FN) and lumbar spine (LS) [38]. All scans were performed by a certified bone densitometry technologist. Sex-specific T-scores for FN and LS-BMD were calculated using the NHANES III reference population [39].

TBS was analyzed using TBS iNsight software version 3.0.0.0 (Med-Imaps, Geneva, Switzerland). Briefly, TBS is a novel grey-level texture measurement, extracted from DXA images, which correlates with 3D parameters of bone micro-architecture, connectivity density, trabecular separation, and trabecular number. For each region of measurement, TBS was evaluated based on grey-level analysis of the DXA images as the slope at the origin of the log-log representation of the experimental variogram. The method of TBS assessment has been described in detail elsewhere [40]

### 2.5. Prevalent Major Osteoporotic Fractures (MOFs)

Fracture events that occurred after the age of 45 years for RS-III and after the age of 55years for RS-I and RS-II were included until the end of 2012. All of the fracture events were reported by general practitioners in the research area by means of computerized systems and by research physicians or trained nurses outside the research area or through hospital records. All reported events were verified by research physicians who independently reviewed and coded the information. Subsequently, a medical expert reviewed all inconsistencies in coded events for final classification.

Fractures were included if they are a component of major osteoporotic fractures (MOF), which includes a fracture of the hip, vertebra (clinical), wrist, or proximal humerus. All of the fractures were coded according to ICD-10 classification. These fractures are the basis for the 10-year absolute fracture risk estimates via FRAX used in multiple large-scale clinical studies [41]. Clinical vertebral fractures were defined as those that came to medical attention when subjects with symptoms (mainly back pain) visited the medical practitioner and the confirmation of fractures occurred on spine radiographs.

### 2.6. Prevalent Vertebral Fractures (VFs)

All if the thoracolumbar radiographs were obtained by a digitalized Fuji FCR system (FUJIFILM Medical Systems, Stanford, CA) according to a standardized protocol described elsewhere [42]. Radiographic vertebral assessment data were available for 3039 subjects with available dAGEs measurements till the end of 2008. Vertebral fractures were classified using vertebral morphometry grading 1 to 3 (OPTASIA-Spina Analyzer) [43]. Because there is doubt as to whether grade 1 (mild) deformities represent true osteoporotic vertebral fractures or a normal variant, we considered grade 2 (moderate) and grade 3 (severe) fractures as radiographic vertebral fractures [44].

Prevalent vertebral fractures (VFs) were then defined as a combination of any vertebral fracture identified on either a radiograph as grade 2 (moderate) or grade 3 (severe) deformity or a clinically reported spine fracture. In this manner, we performed analyses on the most clinically relevant vertebral fractures [44].

### 2.7. Assessment of Covariates

Height (cm) and weight (kg) were measured in the research center with the individuals in standing position wearing indoor clothing without shoes. Body mass index was computed as weight in kilograms divided by height in meters squared (kg/m^2^). The physical activity levels were estimated using an adapted version of the LASA Study Physical Activity Questionnaire [45,46]. Values were recorded in metabolic equivalent of task (MET) hours per week based on questions regarding walking, cycling, sports, gardening, hobbies, and housekeeping. Type 2 diabetes mellitus (T2DM) was defined by combining information on fasting blood glucose levels, antidiabetic medication use, or self-reported medical history. Smoking status was classified as current, former, or never smokers collected through self-report during home interviews. Serum creatinine and serum fasting glucose were measured through automated enzymatic method. Estimated glomerular filtration rate (eGFR) was calculated by the CKD-EPI equation while using serum creatinine concentration, age, and sex data [38].

### 2.8. Statistical Methods

Statistical analyses were performed through IBM SPSS statistics for Windows Version 25.0, Armonk, NY: IBM Corp. The normality of data was determined by the use of histograms and Shapiro–Wilk test. Depending on the distribution of data, it is presented as mean ± SD or median (interquartile range, IQR). The means of continuous variables among groups were compared via the use of Mann–Whitney–Wilcoxon test when a non-normal distribution was assumed or independent samples *t*-test or ANOVA when the variable was normally distributed. Chi-square test was adopted to compare the categorical variables.

For all primary analysis, unless specified otherwise, we used energy-adjusted CML as exposure. For our sensitivity analysis, we also used total dAGE score, MG-H1, and CEL as exposure instead of CML. Potential confounders have been identified and included in the models in order to study the relationship between exposure (CML) and outcomes (BMD/TBS/fractures) based on a common cause approach and on the literature evidence. A consistent approach was used, as follows: Model 1 included age, sex, and RS-cohorts; Model 2 additionally physical activity, dietary quality score, and energy intake, eGFR, type 2 diabetes (yes/no), smoking (never, former, current); Model 3 included model 2 plus BMI and for fracture as an outcome; and, Model 4, including Model 3 plus BMD. Linear regression analysis was used to assess the relationship between dAGEs intake and LS-BMD, FN-BMD and TBS. Logistic regression analysis was performed to assess whether dAGEs intake was associated with the presence of prevalent VFs or prevalent MOFs. A non-linear association was being explored by adding a quadratic term to the original model. However, for all analyses, a linear model had the best fit. Although a linear model fitted best to study these relationships, we also studied dAGEs in quartiles to specifically compare the subjects with highest energy-adjusted AGEs intake.

We tested for interaction terms between CML intake and smoking status and eGFR in the multivariate fully adjusted models. Because AGEs have been repeatedly implicated in pathogenesis of increased fracture risk in T2DM we performed stratified analyses in participants with and without T2DM and both sexes, irrespective of statistically significant interaction.

## 3. Results

### 3.1. Descriptives

Table 1 shows the demographic and clinical characteristics of the participants. The mean age of the population is 66.7 ± 10.5 years, with a BMI of 27.4 ± 4.2 kg/m^2^ and eGFR of 77.9 ± 14.9 mL/min. Our population consisted of 43% males, 12% subjects with T2DM and 12% with a eGFR less than 60 mL/min. Total daily energy intake was 2154.6 ± 682.7 kcal/day with a median physical activity of 41.5 (64.6) MET h/week. 332 participants (8.4%) had a prevalent MOF and 294 (7.5%) had a prevalent VF.

A comparison of subjects with and without T2DM showed that T2DM participants were older (72.0 ± 9.2 vs. 66.2 ± 10.6), have more males (49.7% vs. 42.2%), and higher BMI (29.7 ± 4.8 vs. 27.0 ± 3.9). Subjects with T2DM have significantly higher intake of energy-adjusted MG-H1 (29.1 ± 7.8 vs. 28.3 ± 7.7 mg/day) and CEL (2.57 ± 0.97 vs. 2.39 ± 0.86 mg/day) and no difference in CML intake (2.49 ± 0.93 vs. 2.41 ± 0.86 mg/day), although a trend towards higher intake, but they have lower energy intake (2045 ± 690 vs. 2170 ± 680 kcal/day) and physical activity (31.6 (54.9) vs. 42.7 (65.8) MET h/week) than subjects without T2DM. Despite having higher BMD, subjects with T2DM have not lower prevalence of fractures than those without T2DM (see Table 1).

### 3.2. Linear Regression Analysis of Energy-Adjusted CML Intake (CML) with Bone Mineral Density (BMD) and Trabecular Bone Score (TBS)

Table 2 show the results of the linear regression analysis describing the association of dietary CML intake with BMD and TBS. In linear regression models for all the covariates (as mentioned), we did not observe any associations between CML intake and FN-BMD (β = −0.006; *p* = 0.70), CML intake and LS-BMD (β = −0.013; *p* = 0.38) and CML intake and TBS (β = −0.015; *p* = 0.48). There was also no interaction by sexes, diabetes, and smoking status and eGFR.

### 3.3. Logistic Regression Analysis for the Association between Energy-Adjusted CML Intake (CML) and Prevalence of Fractures

#### 3.3.1. Major Osteoporotic Fractures (MOFs)

In Model 3 (fully adjusted model), the odds ratio OR (95% CI, *p*-value) of the CML intake for MOFs was 1.12(0.98–1.27, *p* = 0.10). After additional adjustment for FN-BMD, there is minimal attenuation of the OR to 1.11(0.98–1.26, *p* = 0.12) (Table 3). The prevalence of MOFs was 11.1% in females and 4.4% in males. Stratification on the basis of sex showed that CML intake is not significantly associated with higher risk of fracture in females (OR = 1.10 (0.94–1.28, *p* = 0.24) and males (OR = 1.17 (0.91–1.49, *p* = 0.28). The prevalence of MOFs is 9.4% in subjects without T2DM and 9.9% in those with T2DM. Stratification on the basis of diabetes status showed that the association between CML intake and MOFs in subjects with T2DM (OR = 1.38 (0.97–1.98, *p* = 0.08) is marginally non-significant although the effect size is quite high. No significant association was found in those without T2DM (OR = 1.08 (0.93–1.24, *p* = 0.31) (Figure 2/Appendix A).

#### 3.3.2. Vertebral Fractures (VFs)

In Model 3 (fully adjusted model), the odds ratio OR (95% CI, *p*-value) of the CML intake for VFs was 1.16(1.02–1.32, *p* = 0.02). After additional adjustment for FN-BMD, there was no change in the OR: 1.16(1.02–1.31, *p* = 0.025) (Table 3). The prevalence of VFs was 7.1% in females and 8.0% in males. Stratification by sex showed that CML intake was associated with significantly higher risk of fracture in females (OR = 1.22 (1.02–1.45, *p* = 0.03), but not in males (OR = 1.11 (0.92–1.34, *p* = 0.28). Prevalence of VFs is 7.5% in subjects without T2DM and 7.4% in those with T2DM. Stratification by diabetes status showed that CML intake was associated with significantly higher risk of fracture in subjects without T2DM (OR = 1.16 (1.01–1.33, *p* = 0.03), but not in those with T2DM (OR = 1.10 (0.76–1.61, *p* = 0.61) (Figure 2/Appendix A).

### 3.4. Logistic Regression Analysis for the Association between Top Food Categories Contributing to CML and Prevalence of Fractures

The food items of the FFQ were identified into 25 major food categories. The contribution of individual food categories to daily CML intake was calculated by dividing the CML intake derived from that food category by the total CML intake daily. Figure 3 shows the top 10 food categories contributing to more than 85% of daily dietary CML intake, namely: sweets, whole grains, unprocessed meat, refined grains, processed meat, nuts, pulses, fish and seafood, and yogurt.

Table 4 shows the top 10 categories independently and a comparison with top six combined and top 10 combined in terms of odds contributing to MOFs and VFs in fully adjusted models.

#### 3.4.1. Major Osteoporotic Fractures (MOFs)

Comparing effect size from individual categories showed that processed meat, yogurt, milk, and grains are a major contributor to the observed odds for MOFs, if any. A comparison of the top six AGEs food categories with top 10 showed a very slight difference in the effect size measured as odds ratio (top six OR = 1.09 (0.97–1.22) vs. top 10 OR = 1.11 (0.98–1.27)).

#### 3.4.2. Vertebral Fractures (VFs)

Comparing effect size from individual categories showed that unprocessed and processed meat, yogurt, milk, and sweets are a major contributor to the association of total CML intake with the presence of VFs. A comparison of the top six AGEs food categories with top 10 did not show any difference in the effect size measured as odds ratio (top six OR = 1.15 (1.03–1.29) vs. top 10 OR = 1.15 (1.01–1.30)).

### 3.5. Subgroup and Sensitivity Analysis

We observed neither an association for MOFs with respect to total dAGEs score [1.07 (0.88–1.30)], MG-H1 [1.00 (0.98–1.02)], or CEL [1.00 (0.88–1.15)] as continuous variables nor when comparing the participants in different quartiles (Q1 to Q4).

We observed no association for VFs with respect to total dAGEs score [1.08 (0.89–1.31), MG-H1 [1.00 (0.98–1.01)], and CEL [1.02 (0.90–1.17)] as continuous variables; however, when comparing the participants with values in bottom three quartiles (Q1 to Q3) of total dAGEs to top quartile [Q4: 1.40 (1.00–1.96)], the odds for VFs were significantly higher. We observed no associations for both MG-H1 and CEL with VFs when comparing different quartiles to each other (Appendix A).

## 4. Discussion

This study investigated the association of dietary intake of AGEs, in the form of CML, estimated from a FFQ and dAGEs database with bone health parameters in a cross-sectional way. We observed a linear positive association between CML intake and VFs independent of BMD, but not between CML intake and MOFs, even though there was a similar trend. High consumption of CML had no relation to BMD in our cohort.

Higher CML intake was associated with higher prevalence of VFs in our cohort. A recent study showed that high AGEs diet led to increased total fluorescent AGEs accumulation in cortical and cancellous vertebral bone, inferior vertebral microstructure, and mechanical properties primarily in six-month old female mice, but not in six-month old male mice and mice that were fed a low AGEs diet [28]. Their conclusion that young female mice, without diabetes or overweight, seem to be at increased risk of developing vertebral fracture by high AGEs diet seems to be consistent with our findings of 16% increased risk of VFs with high CML intake in females and subjects without diabetes and.

High CML intake showed a similar trend towards higher prevalence of MOFs in our cohort, but the results were not significant. Studies on animals consuming diets rich in AGEs, such as bread crusts (CML) [47] or high fat showed an increased accumulation of AGEs in tibia and reduced mechanical functioning of cortical bone [27,48]. In contrast, Karim et al. showed a higher correlation between pentosidine, (a cross-linking AGE) and total fluorescent AGEs in human cancellous bone specimens compared to cortical bone specimens (*n* = 170). Moreover, in vitro glycation of a subset of bone specimens (*n* = 28) revealed increased AGEs accumulation in cancellous versus cortical bone, owing to higher surface to volume ratio of cancellous bone [49]. Whether there is a difference in AGEs accumulation in cortical and cancellous bone based on the type of AGEs consumed still needs to be explored.

We did not observe any association between CML intake and BMD or TBS as a surrogate of bone strength. Interestingly, skin AGEs have been associated with measures of bone strength other than BMD [1,2,3]. Whether dietary AGEs or AGEs measured in tissues have a relationship to microarchitecture using sophisticated techniques, such as high-resolution Quantitative Computed Tomography, would be interesting for future research.

Various studies investigated the effect of Mediterranean diet on bone health and reported a positive association [50,51,52] or no association [53,54]. A traditional Mediterranean diet is characterized by low intake of meat and meat products, moderate intake of dairy products, and high amounts of fruits and vegetables (rich in antioxidants and polyphenols), fish, and cereals in unprocessed form [55]. Adherence to the Mediterranean diet has recently been associated with lower hip fracture incidence in a meta-analysis, including 140,775 middle-age subjects [56]. Interestingly, Sanchez et al. reported that higher adherence to Mediterranean diet is inversely associated with skin AGEs accumulation in 2646 middle-aged subjects [57,58]. Similarly, a positive association between the dietary intake of meat and meat products and skin AGEs accumulation was found [59]. Lastly, multiple clinical trials using polyphenols showed reduced AGEs-induced inflammation in tissues, including bone, particularly in subjects with T2DM [60,61]. In conclusion, one of the underlying mechanisms of Mediterranean diet in improving bone health could be related to reduced AGEs intake and the inhibition of AGEs by antioxidants and polyphenols.

A major contribution of dietary CML more than 70% came from following food categories: sweets, milk, whole and refined grains, processed and unprocessed meat. Additionally, the observed fracture risk, although not very strong, could be explained to a large extent through these top contributors to CML independent of BMD. Higher candy consumption has previously been associated with low BMD [62]. High milk consumption (≥3 glasses per day) has been associated with higher hazards of any and hip fracture in women (*N* = 61,433), but not in men after a median follow-up of 22-years [63]. A potential explanation of increased fracture risk that is associated with high milk consumption is its high D-galactose content. The administration of D-galactose is an established animal model of premature aging, partially through its effect on the accumulation of AGEs [64]. Meat intake has been variably associated with negative bone health, for example, by interfering with calcium metabolism by increasing acid load and as a consequence of high sodium content in processed meat [65]. It is important to realize, in this respect, that the contribution to body’s AGEs pool is owed not only to crude AGEs content in a food item, but also to the preparation techniques, such as dry heat, long cooking time, and high temperature, which accelerate AGEs formation. In the current study, we still could not establish that the observed effect on fracture risk comes solely from AGEs and not from the other constituents of these food categories. Future studies on the tissue AGEs accumulation with routine high AGEs food categories, processed in different ways, would provide interesting insights into the pathophysiological mechanisms.

CML served as a representative of dAGEs intake in relation to bone health in our study due to the following reasons: Firstly, CML is among the best studied AGEs [66], not only in all of the dAGE databases, but also in bone research [67,68] while MG-H1 and CEL have hardly been studied in relation to bone (disregarding pentosidine, an AGE that is not yet a part of any dAGEs database). Secondly, CML is a very stable and a relatively inert AGE that is expected to be least influenced by the processing and cooking techniques. Lastly, the correlation among different AGEs in food items, namely CML and CEL (*r* = 0.79), CML and MG-H1 (*r* = 0.78), and CEL and MG-H1 (*r* = 0.90), is quite strong.

There are several strengths of our study. De AGEs database used to estimate dAGEs used a state-of-the-art ultra-performance liquid chromatography/tandem mass spectrometry (UPLC/MS/MS), which is superior to ELISA [31]. The FFQ and dAGEs database have both been primarily designed for Dutch population, which points to a reliable representation of dAGEs. Our cohort was entrenched from a Rotterdam study that is a well-characterized population with a consistent way of data collection.

Several limitations merit further discussion. AGEs are such a heterogeneous group of compounds that the information on the dietary content of a few AGEs may not be sufficient for generalization. Our population based FFQ did not contain information on cooking time, moisture, and temperatures used. Lastly, the quantity, type, and form of AGEs (free adducts vs. protein-bound) in food could affect their bioavailability and eventual contribution to the circulating and tissue pool. These limitations could have led to an over- or underestimation of our findings. We did not correct for multiple testing due to exploratory character of our findings and strong correlation between different AGEs and between different types of fractures, but we observed a consistent trend in all subcategories. To finish, we did not use medications that could influence bone health, as a covariate. Importantly, our results call for replication in independent cohorts in the future.

## 5. Conclusions

This study demonstrated a positive association between dCML intake and prevalence of VFs and a similar but non-significant trend for prevalent MOFs, independent of BMD. Dietary AGEs intake showed no association with BMD and TBS. The observed effect of dCML intake on VFs came primarily from the top six food categories, namely, sweets, milk, grains, and meat. The replication of our findings in independent cohorts will be needed and could lead to potential interesting public health consequences. Future studies with a longitudinal design are mandatory. Additionally, short clinical trials focusing on better quantification of AGEs, by using specially designed questionnaires, including food preparation conditions, could provide interesting insights into the causal mechanisms underlying dAGEs and tissue pathology.

## Figures and Tables

**Figure 1 nutrients-12-02377-f001:**
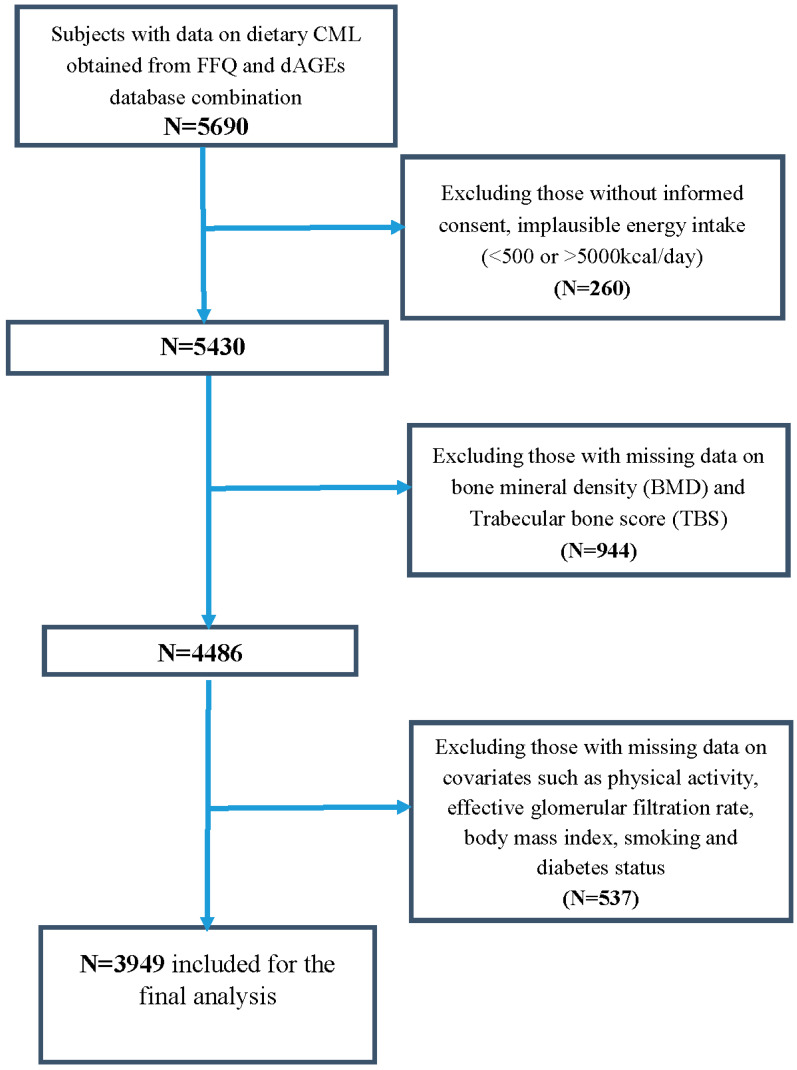
Flowchart of participant inclusion from the Rotterdam Study. CML, Carboxymethyllysine; FFQ, food frequency questionnaire; dAGEs, dietary advanced glycation end-products.

**Figure 2 nutrients-12-02377-f002:**
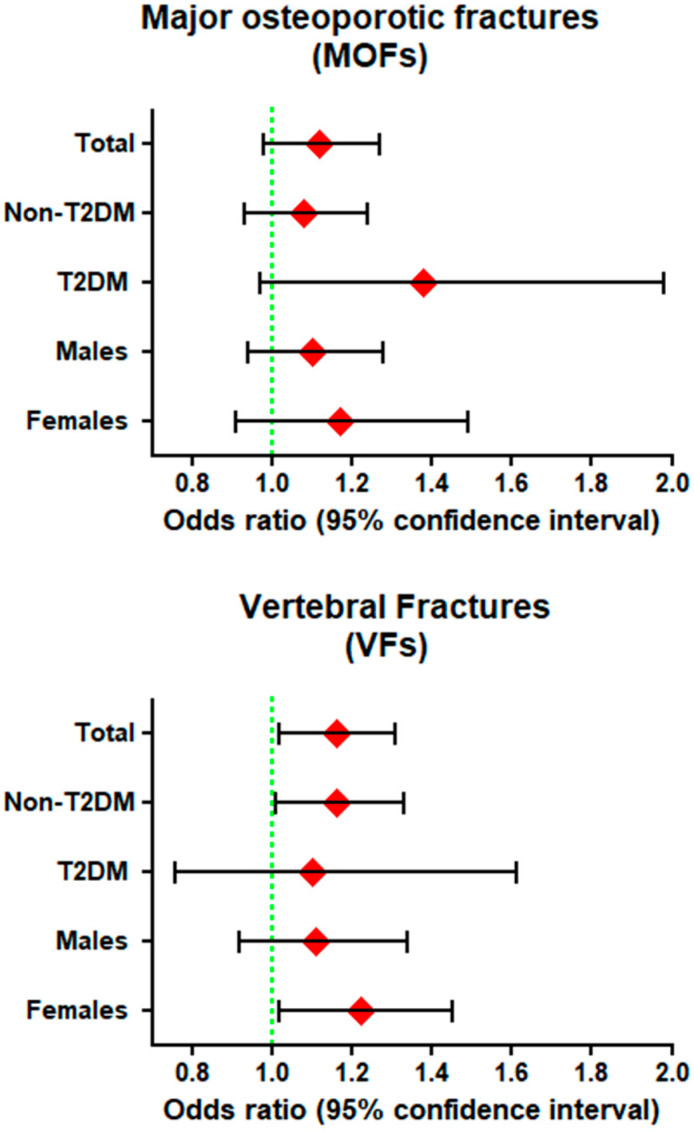
Odds ratio (ORs) of energy-adjusted CML for MOF and VFs in non-type 2 diabetics (non-Table 2. DM), diabetics (T2DM), females and males in *N* = 3949 with complete data on all covariates.

**Figure 3 nutrients-12-02377-f003:**
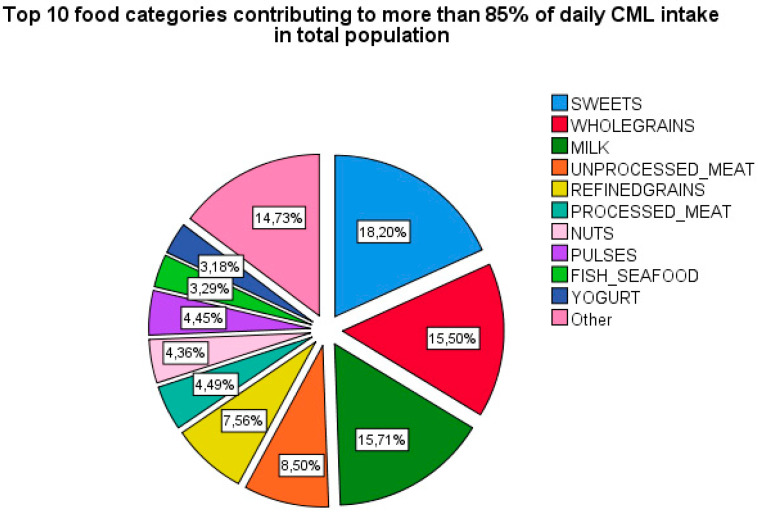
Pie chart showing top food categories contributing to dietary CML intake in.

**Table 1 nutrients-12-02377-t001:** Demographic and clinical characteristics of total participants with complete data and a comparison between subjects with (T2DM) and without type 2 diabetes mellitus (non-T2DM).

	Total Participants (*N* = 3949)	T2DM (*n* = 473) 12%	Non-T2DM (*n* = 3476) 88%
CML (mg/day, energy adjusted)	2.42 ± 0.88	2.49 ± 0.93	2.41 ± 0.86
MGH1 (mg/day, energy adjusted)	28.4 ± 7.73	29.1 ±7.8 *	28.3 ± 7.7
CEL (mg/day, energy adjusted)	2.42 ± 0.87	2.57 ± 0.97 *	2.39 ± 0.86
Age (years)	66.7 ± 10.5	72.0 ± 9.2 *	66.2 ± 10.6
Males, *n* (%)	1703 (43%)	235 (49.7%) *	1469 (42.2%)
BMI (kg/m^2^)	27.4 ± 4.2	29.7 ± 4.8 *	27.0 ± 3.97
**eGFR (mL/min per 1.73 m^2^)**	77.9 ± 14.9	77.6 (22.9) *	79.5 (19.6) *
eGFR < 60, *n* (%)	466 (12%)	92 (19.5%) *	378 (10.9%)
Never smokers, (%)	32%	27.7%	33.1%
Ex-smokers, (%)	52%	59.0%	50.8%
Current smokers, (%)	16%	13.3%	16.1%
Total energy intake, kcal/day	2154 ± 683	2045 ± 690 *	2169 ± 680
Fat intake, g/d	77.9 ± 35.4	74.2 ± 35.1	77.9 ± 34.7
carbohydrate intake, g/day	243.0 ± 87.0	229.6 ± 87.8 *	245.1 ± 85.3
protein intake, g/day	82.6 ± 26.2	81.6 ± 27.9	82.7 ± 25.5
Physical activity (MET hours/week)	41.5 (64.6)	31.6 (54.9) *	42.7 (65.8)
Major osteoporotic fractures, *n* (%)	334 (8.5%)	42 (9%)	292 (8.4%)
Vertebral fractures, *n* (%)	296 (7.5%)	261 (7.5%)	35 (7.4%)
Femoral neck BMD, g/cm^2^	0.901 ± 0.137	0.916 ± 0.142 *	0.899 ±0.136
Lumbar spine BMD, g/cm^2^	1.140 ± 0.217	1.194 ± 0.215 *	1.132 ± 0.216
TBS (unitless)	1.311 ± 0.101	1.297 ± 0.107	1.313 ± 0.101

Carboxymethyllysine, CML; Carboxyethyllysine, CEL; Methylglyoxal-derived hydroimidazolones, MG-H1; Dietary advanced glycation end-products, dAGE; Body mass index, BMI; effective glomerular filtration rate, eGFR; Bone mineral density, BMD; trabecular bone score, TBS; major osteoporotic fractures, MOFs; Vertebral fractures, VFs; Data are presented as mean ± SD, median (interquartile range) and number (%). * Represents a *p*-value < 0.05 between subjects with T2DM and non-T2DM.

**Table 2 nutrients-12-02377-t002:** Association of energy-adjusted carboxymethyllysine (CML) with femoral neck and lumbar spine bone mineral density (BMD) and trabecular bone score (TBS) in the whole population (*N* = 3949).

Outcome	Standardized Coefficient β (*p*-Value)		
	Model 1	Model 2	Model 3
Femoral Neck BMD	0.000 (0.97)	−0.001 (0.73)	−0.006 (0.70)
Lumbar spine BMD	−0.012 (0.41)	−0.012 (0.42)	−0.013 (0.38)
Trabecular bone score	0.003 (0.89)	−0.008 (0.73)	−0.015 (0.48)

Model 1: Energy-adjusted CML + age + sex + RS-cohorts. Model 2: Model 1 + physical activity + dietary quality score + total energy intake per day. Model 3: Model 2 + eGFR + diabetes status + smoking status + BMI.

**Table 3 nutrients-12-02377-t003:** Odds ratio of energy-adjusted CML for Major Osteoporotic Fracture (MOF) and vertebral fractures (VFs) in total population with available BMD measurements (*N* = 3949).

	Major Osteoporotic Fractures (MOFs)		Vertebral Fractures (VFs)	
	ORs (95% CI)	*p*-Value	ORs (95% CI)	*p*-Value
N (%)	**334 (8.5%)**		**296 (7.5%)**	
Model 1	1.08 (0.95–1.24)	0.23	1.13 (0.98–1.27)	0.06
Model 2	1.12 (0.98–1.28)	0.10	1.15 (1.01–1.31)	0.02
Model 3	1.12 (0.98–1.27)	0.11	1.16 (1.01–1.31)	0.02
Model 4 (BMD)	1.11 (0.98–1.27)	0.11	1.15 (1.01–1.31)	0.025

Model 1: Energy-adjusted CML + age + sex + RS-cohorts. Model 2: Model 1 + physical activity + dietary quality score + total energy intake per day. Model 3: Model 2 + eGFR + diabetes status + smoking status + BMI. Model 4 Model 3 + femoral neck BMD.

**Table 4 nutrients-12-02377-t004:** Odds ratio of MOFs and VFs for energy-adjusted CML from the top food categories contributing the total CML intake.

*N* = 3949 Fully Adjusted Models	Major Osteoporotic Fractures (MOFs)		Vertebral Fractures (VFs)	
	ORs (95% CI)	*p*-Value	ORs (95% CI)	*p*-Value
Sweets	1.04 (0.80–1.38)	0.77	1.09 (0.83–1.43)	0.52
Whole grains	1.02 (0.90–1.17)	0.73	1.02 (0.90–1.15)	0.78
Milk	1.06 (0.92–1.18)	0.51	1.09 (0.97–1.22)	0.15
Unprocessed meat	1.06 (0.95–1.18)	0.33	1.11 (0.997–1.24)	0.06
Refined grains	1.07 (0.94–1.21)	0.30	1.05 (0.94–1.18)	0.38
Processed meat	1.40 (0.62–3.12)	0.42	1.74 (0.83–3.67)	0.15
Nuts	1.03 (0.29–1.12)	0.10	0.62 (0.32–1.20)	0.15
Pulses	0.87 (0.49–1.55)	0.63	1.05 (0.63–1.77)	0.85
Fish and Seafood	0.40 (0.12–1.37)	0.15	0.52 (0.16–1.71)	0.28
Yogurt	1.21 (0.96–1.54)	0.11	1.14 (0.88–1.48)	0.33
Top 6 combined	1.09 (0.97–1.22)	0.17	1.15 (1.03–1.29)	**0.02**
Top 10 combined	1.11 (0.98–1.27)	0.10	1.15 (1.01–1.30)	**0.03**

Fully adjusted model: Energy-adjusted CML + age + sex + RS-cohorts + physical activity + dietary quality score + total energy intake per day + eGFR + diabetes status + smoking status + BMI.

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
