# Peer review of "Dietary Advanced Glycation End-Products (dAGEs) Intake and Bone Health: A Cross-Sectional Analysis in the Rotterdam Study"

_nutrients, 2020, doi:10.3390/nu12082377_

Round 1
Reviewer 1 Report
This is an interesting manuscript that investigates the association between dietary advanced glycation end-products and bone measures. There are a few questions for the authors:
- Regarding the sentence: “All scans were 178 performed and verified by a trained technician who applied adjustments when necessary.” What kind of adjustments were applied? Could a little more detail be included here?
- What version of the TBS iNsight software was used to calculate TBS?
- Regarding the sentence: “Fractures were included if they are a component of major osteoporotic fractures (MOF) which includes a fracture of the hip, vertebra (clinical), wrist or humerus.” These appear to be the FRAX definition of major osteoporotic fractures, which includes the proximal humerus, but not other regions of the humerus. Were steps taken to only include proximal humerus fractures? Or were all humerus fractures included?
- For T2DM, were there any participants who had T1DM? Were the authors able to determine what type of diabetes each participant had?
- Could the authors provide some examples of what foods are included in the “sweets” category? The other groups are self-explanatory.
Reviewer 2 Report
The Authors investigated the association between dietary AGE (Carboxymethyllysine, CML) obtained from food frequency questionnaires and bone health. Particularly, they considered measurements of bone mineral density, prevalent major osteoporotic fractures and vertebral fractures.
This cross-sectional study is based on a large sample size and the data analysis is well carried out.
The results are original and highlight a positive association between dietary CML intake and prevalence of vertebral fractures. However, dietary AGEs intake was not significantly associated with bone mineral density or the novel index of bone quality named Trabecular Bone Score.
The authors may speculate in their discussion about the association (not investigated) between AGEs and other surrogate of bone strengths, such us Quantitative UltraSound measurements of bone or Quantitative Computed Tomography evaluation (e.g. Clin Cases Miner Bone Metab. 2013;10(3):191-194; J Diabetes Res. 2020;2020:7608964. doi:10.1155/2020/7608964; J Clin Endocrinol Metab. 2020;105(1):dgz036. doi:10.1210/clinem/dgz036).
Moreover, the Authors should specify whether participants were under particular treatments influencing bone health, or alternatively include this point as a limitation of the study if data are not known.
